# Investigations of Top-Level Domain Name Collisions in Blockchain Naming Services

## ABSTRACT

Traditionally, top-level domains (TLDs) are managed by the Internet corporation for assigned names and numbers (ICANN), and the domain names under them are managed by registrars. Against such centralized management, a blockchain naming service (BNS) has been proposed to manage TLDs on blockchains without authority intervention. BNS users can register TLD strings as non-fungible tokens and manage the TLD root zone. However, such decentralized management results in the introduction of a new security issue, BNS TLD name collision, wherein the same TLD is registered in several different BNSs. In this study, we investigated BNS TLD name collisions by analyzing TLDs registered on two BNSs: Handshake and Decentraweb. Specifically, we collected TLDs registered in Handshake and Decentraweb and the associated data, and analyzed the data registration status of BNS TLDs and BNS TLD name collisions. The analysis of 11,595,406 Handshake and 11,889 Decentraweb TLDs revealed 6,973 BNS TLD name collisions. In particular, lastname TLDs, which are intended for use as person names, yielded a large number of registered domain names. In addition, the analysis identified 10 name collisions between the BNS and operational ICANN TLDs. Further, the ICANN TLD candidates under review also had name collisions against the BNS TLDs. Consequently, based on the characteristics of these name collisions and discussions in BNS communities, we considered countermeasures against BNS TLD name collisions. For the further development of BNSs, we believe that it is essential to discuss with the existing Internet communities and coexist with the existing Internet.

## CCS CONCEPTS

• **Security and privacy** → **Network security**; • **Networks** → **Naming and addressing**.

## KEYWORDS

Blockchain Naming Service, Top-Level Domain, Name Collision

**ACM Reference Format:**
Anonymous Author(s). 2024. Investigations of Top-Level Domain Name Collisions in Blockchain Naming Services. In *Proceedings of The Web Conference (WWW '24)*. ACM, New York, NY, USA, 10 pages. https://doi.org/XXXXXXX.XXXXXXX

## 1 INTRODUCTION

To avoid censorship by authorities and operators in centralized systems and services, the development of decentralized systems and services using blockchain has gained momentum. Decentralized services using blockchain have also been proposed for the domain name system (DNS), which is the Internet's backbone. Traditionally, top-level domains (TLDs), such as `.com` and `.net` are managed by the Internet corporation for assigned names and numbers (ICANN), whereas domain names under TLDs are managed by registrars. In contrast to centralized management, a blockchain naming service (BNS) has been proposed, which uses a blockchain to record TLDs and DNS resource records (RRs). This enables the registration and renewal of TLDs and domain names without the intervention of authorities and registrars [3, 8].

A BNS provides functions for users to manage their namespaces on the Internet using a blockchain. BNS users can register strings as non-falsifiable tokens (NFTs), which can then be treated as TLDs (hereafter, referred to as BNS TLDs) and domain names. This facilitates them in managing and operating the namespaces under their control. In addition to the use of BNS TLDs as namespaces on the Internet, several BNSs provide functions that can be used as namespaces on the blockchain by associating them with wallet addresses [7, 18]. However, such a permissionless decentralized management results in the new problem of **BNS TLD name collision**, wherein the same TLD is registered in several different BNSs [36]. This implies that when the same TLD is registered across several different BNSs, the name resolution results for the TLD differ depending on the resolver used. BNS TLD name collisions cause serious security risks such as name resolution to different IP addresses, resulting in communication failures, or name resolution to different wallet addresses, resulting in incorrect cryptocurrency transfers.

The problem of name collisions has also been a concern in the traditional TLDs (hereafter, referred to as ICANN TLDs) [26]. In fact, several cases have been reported wherein original TLDs used within an organization's network collided with newly added ICANN TLDs, causing communication failures [25, 31]. The occurrence of name collisions affect communications and cause serious security issues, such as information leaks, credential theft, and man-in-the-middle attacks [21, 22]. The ICANN implementation plan for the next round of new gTLD applications was received in July 2023, and more new ICANN TLDs will be registered in the future. Consequently, we can predict an increase in name collisions between ICANN and BNS TLDs. In the current Internet space comprising a mix of centralized and decentralized namespace management methods, investigations of and countermeasures for BNS TLD name collisions in advance are important owing to the problem being more complex than with traditional name collisions. Although several studies have reported BNS abuses for malware, botnets, and domain squatting attacks in

the BNS TLD namespace [20, 33, 34, 37], there are no studies on BNS TLD name collisions.

In this study, we focused on Handshake [8] and Decentraweb [3], which are BNSs that allow users to manage and operate TLDs, to investigate BNS TLD name collisions. Specifically, we collected BNS TLDs, DNS RRs, and owner addresses registered in Handshake and Decentraweb, and analyzed the registration status and the existence of BNS TLD name collisions. In addition, we analyzed the existence of name collisions between BNS and ICANN TLDs. This study makes the following contributions:

- We collected and analyzed 11,595,406 Handshake and 11,889 Decentraweb TLDs, and identified 6,973 BNS TLD name collisions between Handshake and Decentraweb. In particular, lastname TLDs, which are intended to be used for person names, exhibited a large registration number of domain names.
- We identified 10 name collisions between BNS and operational ICANN TLDs. Moreover, we revealed that 2 Handshake TLDs and 2 Decentraweb TLDs exhibited name collisions with ICANN TLD candidates under review.
- We showed that 99.9% DNS RRs of Handshake TLDs were configured automatically by a marketplace, indicating a revert to centralised management. In contrast, only 0.04% Decentraweb TLDs had DNS RRs, indicating that their utilization as DNS was not widespread.
- Further, we identified owners who exclusively held 238,420 Handshake and 1,594 Decentraweb TLDs. In particular, these exclusively held Decentraweb TLDs exhibited high name collision rates.
- Finally, we considered countermeasures against BNS TLD name collisions based on the characteristics of these name collisions and discussions in the BNS communities.

## 2 RELATED WORK

Various researchers have analyzed BNS system inadequacies and malicious behaviors in BNSs. Kalodner et al. investigated the use of domain names in Namecoin's `.bit` TLD, which includes the number of registrations, frequency of updates, availability, and content type [33]. This survey revealed that specific users occupy most of the namespace. Casino et al. collected domain names under the TLDs of Namecoin and Emercoin and reported the abuse for malware and phishing by analyzing IP addresses and metadata associated with the domain names, tracking wallet addresses associated with the domain names, and OSINT analysis [20]. ENS has also been analyzed through large-scale data measurement. Xia et al. discovered that wallet addresses are abused for domain squatting, redirection to malicious websites, and fraud [37]. In addition to these traditional DNS security issues, they reported BNS-specific problems that allow the name resolution of ENS domain names, even though they have expired.

All the above studies analyzed BNSs (i.e., Namecoin, Emercoin, and ENS) that handle domain names under specific TLDs (TLD+1), and not BNSs such as Handshake or Decentraweb, which provide decentralized management from the TLDs (root zone). In contrast, Randall et al. analyzed issues in five BNSs including Handshake from the perspective of anti-malware [34]. They indicated that the

**Table 1: Comparison of Namespace Management Methods on the Internet.**

| Method | Root Zone | Management | TLD Examples |
|--------|-----------|------------|--------------|
| DNS | Root Servers | Centralized | `.com`, `.net`, `.org` |
| Alt. Root | User Servers | Hybrid | `.bbs`, `.cyb`, `.chan` |
| BNS | Blockchain | Decentralized | Unrestricted |

characteristics of BNSs make it difficult to control and delete individual names, making traditional countermeasures such as legal intervention against malicious domain names ineffective. As an alternative, they proposed countermeasures for resolvers and proxies used for the name resolution of BNS domain names.

Although several studies have investigated BNS abuse, to the best of our knowledge this is the first study focusing on BNS TLD name collision.

## 3 BACKGROUND

### 3.1 DNS and Alternative Root

A comparison of DNS, alternative root, and BNS, which are methods for managing namespaces on the Internet, is shown in Table 1. The DNS is a system for managing and operating domain names on the Internet, and ICANN TLDs are primarily used for domain names. The ICANN TLD root zone is managed by 13 root servers distributed worldwide. TLDs managed by ICANN are broadly classified into generic TLDs (gTLDs) and country-code TLDs (ccTLDs), which represent countries and regions [32]. New gTLDs are called, reviewed, approved, and registered through the ICANN's New gTLD Program, with 1,930 new gTLD applications received in 2012 [29]. Domain names using ICANN TLDs can be obtained by applying for registration with an ICANN-accredited registrar and being registered in a database maintained by an ICANN-designated registry. Thus, the DNS employs centralized management under the supervision of ICANN.

Alternative roots provide their own namespaces, including TLDs, not under ICANN supervision, by managing and operating their own root servers. OpenNIC, a prime example, provides uncensored name servers and new TLDs can be proposed by community participants [15]. Although they provide a democratic and non-state alternative to ICANN TLDs, a part of centralized management remains, as the adoption of a new TLD must satisfy the requirements and be approved by the community.

### 3.2 BNS

The BNS allows TLD owners, root zones, and domain names to be managed and verified on the blockchain. BNS users can register any string as an NFT, treat it as a TLD, and manage the root zone and DNS RRs in the blockchain. New TLDs can be registered by paying the auction bid price of the TLD string, or a certain amount of money in cryptographic assets. This registration procedure is defined by a permission-less algorithm, primarily a smart contract without intervention authorities or other third parties.

Namecoin, which has been in operation since 2011, uses a blockchain that extends the Bitcoin fork to store JSON data, including domain names and DNS RRs for `.bit` TLD on blockchains [13]. Emercoin,

whose design was inspired by Namecoin, manages domain names and DNS RRs using `.coin` and `.emc`, `.lib`, and other TLDs with data in name and value formats using EmerDNS [6]. In addition to setting DNS RRs in BNS TLDs, Ethereum Name Service (ENS) [7] and Unstoppable Domains [18] allow users to set their wallet addresses in BNS TLDs as a namespace on the blockchain. In contrast to the management of domain names under a specific BNS TLD (i.e., TLD+1), such as the BNSs mentioned above, Handshake [8] and Decentraweb [3] allow users to decentralize the management of the BNS TLDs and their root zones. A comparison of the features of the above BNSs is presented in Table 11 in the Appendix.

To investigate name collisions in BNS TLDs, this study focused on Handshake and Decentraweb, which allow users to register and manage the TLD root zone.

*3.2.1 Handshake.* Handshake is a decentralized permission-less naming protocol that allows peers participating in blockchain networks to verify and manage DNS root zones to build an alternative to the existing DNS [8]. Thus, it is an alternative root managed and operated by blockchain. Handshake's blockchain uses Bcoin fork. The full node `hsd` for joining the blockchain network and the client `hnd-cli` for accessing Handshake are all open source [9].

**TLD Registration and Maintenance Costs:** Costs of Handshake TLD registration are determined by auction. Specifically, the user with the highest bid pays the second-highest bid to register a Handshake TLD. Therefore, popular TLDs with many bids tend to be expensive when registering. Whereas TLDs with few bids tend to be inexpensive, and in certain cases, free. After acquiring a Handshake TLD, a transaction fee (mining cost only) must be paid every two years to make a RENEW transaction and renew TLD ownership.

**Restrictions for TLD Registration:** Fundamentally, there are no restrictions on TLD strings. However, strings related to existing ICANN TLDs, the names of companies included in the Alexa Top 100,000 list, and several existing BNS TLDs are pre-reserved and cannot be registered by general users [11].

**Records**[1]: In addition to the DS, NS, and TXT records, Handshake TLD records can be set with the GLUE4 and GLUE6 records, which can set both the domain name and IP address of a name server, or with the SYNTH4 and SYNTH6 records, which can only set the IP address of a name server as on-chain data. Thus, by setting name servers for BNS TLDs, domain names and DNS RRs under the TLDs can be managed as off-chain data in the same manner as in traditional DNS.

**Metadata**[1]: Handshake TLD metadata includes the owner address, expiration date, and auction status. The auction statuses include OPENING, BIDDING, REVEAL, CLOSED, etc. It can be confirmed whether a TLD has an owner via the status.

*3.2.2 Decentraweb.* Decentraweb provides functions to register TLD namespaces and namespaces for Web3 to represent identities and brands [3]. Decentraweb's blockchain uses Ethereum and Polygon, and all clients and SDKs that interact with Decentraweb's smart contracts are open-source [4]. Decentraweb's TLD is issued as an NFT compliant with the ERC-721 standard and is intended to

be used as a domain name, as well as wallet addresses and social networking service (SNS) accounts.

**TLD Registration and Maintenance Costs:** The cost of Decentraweb TLD registration increases with the number of years of registration. The minimum cost is approximately 50 USD (as of August 2023). In addition, if owners continue to hold their TLDs after the registration period, owners must pay a renewal fee based on the number of years.

**Restrictions for TLD Registration:** Fundamentally, there are no restrictions on TLD strings. However, strings related to existing ICANN TLDs and the names of companies included in the Alexa Top 10,000 list, brand names, and crypto project names are pre-reserved and cannot be registered by general users [5].

**Records**[1]: Decentraweb TLD records can be set with DNS RRs (A, AAAA, CNAME, MX, TXT), resolver addresses, wallet addresses, IPFS content, and text data as on-chain data. The text data includes contact information (email addresses and phone numbers), URLs, and SNS account IDs such as Twitter/X and Github. However, a resolver address must be set in advance to set other records.

**Metadata**[1]: Decentraweb TLD metadata include the owner address, expiration date, and number of registered domain names under the TLD (also referred to as sub- or second-level domains; SLDs).

## 3.3 Name Collision

Name collisions have been a concern in the past [26]. Many new gTLDs have been registered by the ICANN since 2012, and problems, such as communication failure with intended recipients or communication with unintended recipients, have emerged. For example, there was a case of communication failure owing to a name collision between the original TLD used in the organization's network and the new ICANN TLD [25, 31]. In another case, the `.biz` TLD, an alternative root used before the ICANN approval, was later approved, and the name resolution results (IP addresses) of domain names using the `.biz` TLD was changed [24]. Name collisions result in serious security issues such as information leaks, credential theft, and man-in-the-middle attacks. This is attributed to their impact on existing mechanisms such as certificates of domain names, same-origin policies on the Web, and the web proxy auto-discovery protocol [21, 22]. Because the ICANN's implementation plan for the next round of new gTLDs applications was received in July 2023 [35], we expect name collisions with new ICANN TLDs to recur in the future.

Name collision problems also occur in BNS TLDs. For example, `.coin` TLD, which was previously provided by Emercoin, also contains Unstoppable Domains, thereby resulting in name collisions. Unstoppable Domains later identified this and stopped the support for `.coin` TLD [36]. Other litigation issues regarding the name collision of `.wallet` TLD between Unstoppable Domains and Handshake have also been reported [19]. Several BNSs connect TLDs with wallet addresses; therefore, name collisions in BNS TLDs result in a serious risk of misdirected cryptocurrency transfers because the BNS name is resolved to a different wallet address owing to the difference in resolvers, even if the names are the same. Because BNSs employ decentralized and permission-less management, and there exists no centralized organization or coordinating forums

---

[1]In this study, among data associated with a BNS TLD, data to which users can set arbitrary values are referred to as records, and data other than records are referred to as metadata.

such as ICANN, name collisions are likely to occur in the future. In addition, based on the next applications for the ICANN's New gTLD Program, name collisions between BNS and new ICANN TLDs are expected.

### 3.4 Countermeasure Against Name Collisions

Traditionally, ICANN has adopted the name collision occurrence management framework to address name collisions [28]. This framework requires registry operators to take technical measures, referred to as controlled interruption (CI), which aim to mitigate name collisions. Specifically, upon receiving a report of a name collision from ICANN, registry operators must respond with IPv4 loopback address `127.0.53.53` as resolution results for domain names with name collisions. Through the CI response (`127.0.53.53`), registry operators are encouraged to alert system administrators and users of name collision problems.

The ICANN also publishes guidelines [27] for name collision identification and mitigation for IT professionals. Further, it provides extensive information on the causes and potential effects, and offers guidance on how and when to launch mitigation efforts. The guidelines explain how name collisions can be identified by monitoring network traffic to authoritative DNS servers to identify all original TLDs and CI responses and then matching them with the latest TLD namespace information provided by ICANN [23]. The method for mitigating name collisions involves listing relevant devices and systems that use the original TLDs, informing users, and replacing the original TLDs with ICANN TLDs. However, if the replacement is challenging, the guidelines state that name collisions can be avoided by completely isolating the systems using the original TLDs without connecting them to the Internet.

## 4 INVESTIGATION METHOD

### 4.1 Data Collection

*4.1.1 Collection of BNS TLDs.* Handshake TLDs can be collected by analyzing transaction data on the blockchain. Specifically, after synchronizing the blockchain data using the full node `hsd`, this study performed the following analysis on blocks from block height 0 to a specific block height using the client `hsd-cli`.

(1) Obtain the block data for a specific block height using the `getblockbyheight` command.
(2) Extract transaction data contained in the block data.
(3) Extract hash values of domain names contained in the transaction data.
(4) Obtain the strings of the domain names from the hash value using the `getnamebyhash` command.

Decentraweb TLDs can be collected from the contract addresses of Ethereum and Polygon blockchains[1], which are responsible for publishing a TLD as an NFT. There are two methods for collecting TLDs using these contract addresses: analyzing on-chain data or using third-party services. The former method requires additional processing, such as decoding the contract event logs, to extract TLDs. On contrast, the latter method can easily collect TLDs using APIs provided by third-party services. Therefore, in this study,

---

[1]The address of Ethereum is `0x3eAf3D0E21F452adF632744B5608e6C02e88827A` and that of Polygon is `0x5792c3534f6231b1e019740C6079233b3d021Dfe`.

the latter method using the third-party service NFTPort [14] was adopted to collect Decentraweb TLDs.

*4.1.2 Collection of Records and Metadata.* We used the `hsd` and `hsd-cli` commands to collect records and metadata for Handshake TLDs in the same manner as the abovementioned TLD collection. DNS RRs can be collected by the `getnameresource` command and metadata can be collected by the `getnameinfo` and `gettxout` commands.

We used SDK and NFTPort to collect records and metadata for Decentraweb TLDs. The SDK was used to collect all records, whereas NFTPort was used to collect metadata. Note that we only collected BTC and ETH wallet addresses in this study, although multiple types of wallet addresses can be set for Decentraweb TLDs.

### 4.2 Data Analysis

We identified the presence or absence of BNS TLD name collisions, and analyzed DNS RRs and owner addresses of BNS TLDs.

*4.2.1 Analysis of BNS TLD Name Collisions.* We identified the BNS TLD name collisions (registrations of the same TLD) between Handshake and Decentraweb. In addition, we analyzed the BNS TLDs with name collisions for domain squatting, string length, character type, registration date, and SLD registration. In an analysis of domain squatting, we identified TLDs that exactly matched the names of famous organizations, corporations, brands, and web services. Domain squatting is the practice of registering and/or using a domain name with malicious intent to profit from the rights of a trademark belonging to another person. Therefore, we extracted registrable domain strings (strings excluding the public suffix) of domain names collected from the Tranco top 1M list [17] and from the websites of companies with more than 1,000 employees listed on D&B Hoovers [2]. Consequently, the BNS TLDs that matched them exactly were analyzed.

*4.2.2 Analysis of ICANN TLD Name Collisions.* We investigated the impact of name collisions by identifying the existence of name collisions between BNS and operational ICANN TLDs [23] and analyzing the number of domain names under the TLDs with name collisions and the countermeasures applied to them. In addition, we identified TLDs with a high likelihood of name collisions by analyzing the presence or absence of name collisions between BNS and TLDs applied to the New gTLD Program.

*4.2.3 Analysis DNS Resource Records.* We analyzed the settings of DNS RRs of BNS TLDs with name collisions. Specifically, we analyzed the number of TLDs with each DNS RR, IP address, and domain name set in the records.

*4.2.4 Analysis of Owner Addresses.* We analyzed the owner addresses of BNS TLDs with name collisions. Specifically, we analyzed the number of unique owner addresses and TLDs held by each owner address.

## 5 INVESTIGATION RESULTS

### 5.1 Collection Results of BNS TLDs and Records

We collected BNS TLDs, records, and metadata between July 27, 2023, and August 10, 2023. Handshake data were collected from

**Table 2: Collection Results of BNS TLDs and Records.**

|  | Handshake | Decentraweb |
|---|---|---|
| Total Number of BNS TLDs | 11,595,404 | 11,889 |
| # of Valid BNS TLDs | 11,042,189 | 8,134 |
| # of TLDs with Resolver Address | N/A | 3,136 |
| # of TLDs with DNS RRs | 8,989,300 | 3 |
| # of TLDs with Other Records | N/A | 74 |

**Table 3: Breakdown of Number of Other Records Set in Decentraweb TLDs.**

| Record | Number |
|---|---|
| Wallet Address | 67 |
| IPFS URL | 8 |
| Mail Address | 7 |
| Phone Number | 1 |
| URL | 1 |
| SNS Account ID | 1 |
| Keyword | 1 |

block heights of 0—184,000 (12:38, August 4, 2023). The collection results are presented in Table 2. In this study, we collected 11,595,406 Handshake and 11,889 Decentraweb TLDs and obtained 11,607,295 BNS TLDs. Here, 553,217 (4.8%) Handshake TLDs and 3,755 (31.6%) Decentraweb TLDs were invalid TLDs owing to expiration or other reasons; therefore, we excluded them from further analysis. In addition, we could not collect the records of 4,998 (61.4%) valid Decentraweb TLDs, owing to the lack of resolver address settings. Several more TLDs were registered in Handshake than in Decentraweb, possibly because of registration and maintenance costs. Most Handshake TLDs were registered inexpensively, with approximately 80% registered for free, as shown in Fig. 2 in the Appendix. In contrast, Decentraweb TLDs require the payment of registration and maintenance fees based on the number of years. Therefore, we considered that the financial costs affected the number of TLD registrations.

Handshake and Decentraweb TLDs with DNS RRs were 8,989,312 (81.4%) and 3 (0.04%), respectively. When a Handshake TLD is registered through the marketplace Namebase, the NS, GLUE4, and DS records are automatically set for the TLD after the auction. Therefore, the DNS RR setting rate for Handshake TLDs was high. However, Decentraweb users must set DNS RRs to a TLD after setting the resolver address. Owing to the transaction fees required for each of these settings, the DNS RR setting rate for the Decentraweb TLDs was low.

A breakdown of the number of other records set in the 74 (0.9%) Decentraweb TLDs is presented in Table 3. There were 67 (0.8%) wallet address registrations, which is more than the number of DNS RRs. This indicated that Decentraweb TLDs were used more as wallet address aliases than as alternative DNS. However, only 78 TLDs (1.0We considered that most Decentraweb users were interested only in holding TLDs and not in using them. This suggests that, as with traditional domain squatting, many users only registered BNS TLDs that were likely to become expensive in advance.

## 5.2 Analysis Results of BNS TLD Name Collisions

We analyzed 11,061,354 Handshake and 8,134 Decentraweb TLDs and identified 6,973 name collisions. The number of name collision

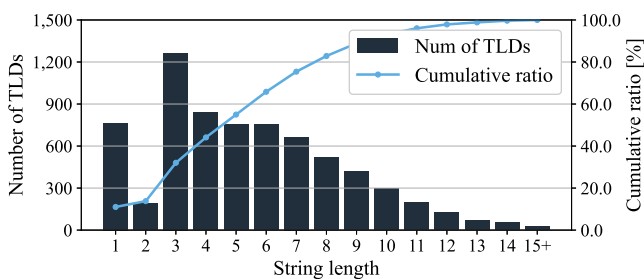

**Figure 1: Distribution of String Length of BNS TLD with Name Collisions**

TLDs according to the string length is shown in Fig. 1. As evident, the shorter the string, the more name collisions that occurred. Name collisions for 3-character TLDs were the most common (1,264), and those for 5-character or less TLDs accounted for over 50% of the total. Note that Decentraweb TLDs are restricted to registering the ASCII characters of two characters or less as TLDs, and only non-ASCII characters (i.e., internationalized domain names (IDNs) and emojis) are allowed for one- and two-character TLDs. This results in the highest number of name collisions for three-character TLDs. Alphabetic characters accounted for the largest number of name collisions at 5,239 (75.1%), followed by emojis or IDNs at 890 (12.8%), numbers at 360 (5.2%), combinations of emojis and alphanumeric characters at 288 (4.1%), and alphanumeric characters at 196 (2.8%). Because there are more types of emojis than alphanumeric characters, with over 3,000 types, there were frequent name conflicts caused by a single emoji rather than a combination of character types, such as alphanumeric characters. By contrast, TLDs combining alphanumeric characters and emojis had fewer name collisions, suggesting that combining multiple character types is effective in avoiding BNS TLD name collisions.

Next, the analysis of TLD registration dates revealed that highest number of Handshake TLD registrations was 5,610 (80.5%) from February 2020 to January 2021. Whereas, that of Decentraweb TLD registrations was 6,118 (92.3%) from October to December 2021. The distribution of TLD registration dates is shown in Fig. 3 in the Appendix. Handshake and Decentraweb made their first transactions on 4 February, 2020, and 8 October, 2021, respectively, and both services exhibited numerous name collisions in BNS TLDs registered several months after the service launch. Since 2023, 33 Handshake and 241 Decentraweb TLDs experienced name collisions, indicating that BNS TLD name collisions occurred continuously.

The top 10 Decentraweb TLDs with the highest number of SLD registrations are shown in Table 4. Seven of the top ten were lastname TLDs for person names, all of which had more than 3,000 SLDs registered. Many SLDs were registered in the form of "firstname.lastname". Although the number of registrations of Handshake SLDs cannot be aggregated, there is a Handshake domain name registration service for person names [16][2]. Thus, based on a large number of people with the same name in the real

---

[2]Although the registration status of person domain names using Handshake TLDs can also be analyzed manually from the service GUI, this analysis was not conducted in this study owing to the load on the service.

**Table 4: Top 10 Decentraweb TLDs with Name Collisions with Highest Number of Registered SLDs.**

| TLD | # of SLDs |
|---|---|
| .contractor | 5,576 |
| .roofing | 5,504 |
| .rock | 4,073 |
| .yang | 3,769 |
| .prince | 3,631 |
| .gonzalez | 3,620 |
| .martinez | 3,612 |
| .rodriguez | 3,609 |
| .hernandez | 3,600 |
| .perez | 3,600 |

**Table 5: Number of Name Collisions Between ICANN New gTLD Program and BNS TLDs.**

| | Handshake | Decentraweb | Total |
|---|---|---|---|
| New gTLD Program | 136 | 44 | 153 |
| Final Status | 130 | 39 | 144 |
|    Delegated (Active) | 8 | 2 | 8 |
|    RA Terminated | 29 | 9 | 35 |
|    Withdrawn | 93 | 28 | 101 |
| Other Status | 6 | 5 | 9 |
|    In Contracting | 0 | 1 | 1 |
|    Applicant Support | 1 | 0 | 1 |
|    On-hold | 1 | 1 | 1 |
|    Will Not Proceed | 0 | 1 | 1 |
|    Not Approved | 4 | 2 | 5 |

world, we can expect that many name collisions will occur in (first-name) SLDs under lastname TLDs, in addition to name collisions in lastname TLDs.

A domain squatting analysis revealed that 3,179 BNS TLDs (45.6% of all BNS TLDs with name collisions) used the same strings as the domain names of famous companies, organizations, brands, and services. The acquisition of TLDs in the expectation of resale has been observed across multiple BNSs, with a tendency towards name collisions. We expect that TLDs containing keywords (buzzwords) related to current trends and social conditions will continue to be acquired across multiple BNSs, resulting in future name collisions.

### 5.3 Analysis Results of ICANN TLD Name Collisions

We identified name collisions with operational ICANN TLDs for 10 Handshake TLDs and 2 Decentraweb TLDs for a total of 10 unique TLDs. Among these, the .music and .kids TLDs were registered in both BNSs. Owing to the name resolution of these TLDs, .music of ICANN and Handshake TLDs responded with 127.0.53.53, implying CI. However, the ICANN .music TLD was subject to CI from Oct 31, 2021, to Jan 28, 2022 [26], which is now operated by a community named DotMusic that has opened the pre-registration of domain names under the .music TLD.

Table 5 presents the analysis results for name collisions with the ICANN TLDs proposed by the New gTLD Program for each review status. The review statuses include "Delegated," "Registry Agreement (RA) Terminated," and "Withdrawn," meaning review completed, and others such as "In Contracting," "Applicant Support," "On-hold," "Will Not Proceed," and "Not Approved" [30]. We identified name collisions with TLDs applied to the New gTLD Program for 136 Handshake and 44 Decentraweb TLDs, for a total of 153 unique TLDs. The delegated TLDs that have been reviewed and are

operational had name collisions in 8 Handshake and 2 Decentraweb TLDs, for a total of 8 unique TLDs (including in 10 name collisions with operational ICANN TLDs, as mentioned above). In addition, TLDs with "In Contracting," "Applicant Support," and "On-hold" status should be aware of name collisions as they may be reviewed and operational in the future. However, 2 Handshakes and 2 Decentraweb TLDs with 3 unique TLDs had already collided with these ICANN TLD candidates.

We investigated the registration dates, number of domain names under the ICANN TLDs, and CI applications for BNS TLDs with name collisions with ICANN TLDs and candidates. We used the data provided by Zonefiles [38] to count the number of domain names under the TLDs. The results are presented in Table 6. The analysis of blockchain transactions for each BNS TLD in Table 6 shows that the first transaction occurred within a few months of the service launch date for almost all BNS TLDs. ICANN TLDs with many domain names such as .ink and .wiki TLDs mainly experienced name collisions with Handshake TLDs. However, the transactions for these Handshake TLDs recorded the claim (C) rather than the auction bid (B). This implies a claim by publishing a DNSSEC ownership proof for a pre-reserved TLD that is considered to have been acquired by the original owner [12]. Although many TLDs acquired through auction bids have only a few registered domain names, which can be assumed to be unused or preproduced, only the .kids TLD had many domain names and was registered in both BNSs. In addition, the .xn-jlq480n2rg, .music, and .kids TLDs were acquired before the ICANN delegation. Only the .music TLD of ICANN and Handshake applied CI (i.e., responded with 127.0.53.53), as mentioned above.

### 5.4 Analysis Results of DNS Resource Records

The results of counting DNS RRs configured for BNS TLDs are shown in Table 7 by all BNS TLDs ("All"), TLDs with name collisions between BNS TLDs ("BNS"), and TLD with name collisions between operational ICANN TLDs ("ICANN"). Regardless of the presence or absence of name collisions, Table 7 shows that there were many NS, GLUE4, and DS records in the Handshake TLDs. This is because these DNS RRs were automatically set for TLDs registered through the marketplace Namebase, as mentioned above. Note that DS records are now automatically set after a specific date; thus, fewer records are set than the NS and GLUE4 records. In contrast, Decentraweb TLDs had few DNS RRs; therefore, their use as an alternative root DNS has not been widespread. We attribute this to the financial cost of the transaction fees mentioned above.

Table 8 presents the results of counting the top five IPv4 addresses set in the DNS RRs in the same manner as the DNS RRs. Although the Handshake TLDs contained a large number of registered IP addresses, the total number of unique IP addresses was 401. Thus, most of the Handshake TLDs had the same IP addresses. Specifically, 44.231.6.183 (99.9%) and 54.214.136.246 (85.5%) were the IP addresses of the name servers operated by Namebase and were the values obtained by the automatic setting mentioned above [10]. The number of unique IP addresses that were likely set by the owners themselves was 276, indicating that the substantial management of BNS TLDs as an alternative root is not decentralized; rather, it reverts to centralization by specific marketplaces.

**Table 6: Name Collisions Between Operational ICANN TLDs and BNS TLDs.**

| | ICANN | | | | Handshake | | | Decentraweb | |
|---|---|---|---|---|---|---|---|---|---|
| gTLD | Application Status* | Delegation Date | # of Domains* | CI** | Bid/ Claim | First TX Date | CI** | First TX Date | CI** |
| .pw | N/A | Jun 30, 2003 | 13,134 | | C | May 27, 2022 | | N/A | N/A |
| .xn--4dbrk0ce | N/A | Jan 14, 2021 | 2 | | B | Oct 22, 2020 | | N/A | N/A |
| .tattoo | Delegated | Nov 14, 2013 | 5,262 | | C | Aug 12, 2022 | | N/A | N/A |
| .wiki | Delegated | Feb 19, 2014 | 54,133 | | C | May 5, 2022 | | N/A | N/A |
| .ink | Delegated | Mar 11, 2014 | 76,157 | | C | May 10, 2022 | | N/A | N/A |
| .gay | Delegated | Aug 9, 2019 | 26,021 | | C | May 5, 2022 | | N/A | N/A |
| .xn--jlq480n2rg | Delegated | Jun 2, 2020 | 2 | | B | May 13, 2020 | | N/A | N/A |
| .xn--cckwcxetd | Delegated | Jun 2, 2020 | 2 | | B | Feb 6, 2021 | | N/A | N/A |
| .music | Delegated | Oct 29, 2021 | 7 | ✓ | B | May 2, 2020 | ✓ | Oct 10, 2021 | |
| .kids | Delegated | Apr 4, 2022 | 3,902 | | B | Jul 31, 2020 | | Oct 16, 2021 | |
| .hotel | On-hold | N/A | N/A | N/A | B | Dec 10, 2020 | | Oct 10, 2021 | |
| .merck | Contracting | N/A | N/A | N/A | | N/A | N/A | Oct 12, 2021 | |
| .idn | App. Support | N/A | N/A | N/A | B | Apr 18, 2020 | | May 9, 2022 | |

\* As of August 31, 2023, \*\* Controlled Interruption

**Table 7: Top 5 DNS Resource Records Set in BNS TLDs.**

| Handshake | | | | Decentraweb | | | |
|---|---|---|---|---|---|---|---|
| Record | All | BNS | ICANN | Record | All | BNS | ICANN |
| NS | 8,912,639 | 5,699 | 6 | A | 2 | 2 | 0 |
| GLUE4 | 8,864,326 | 5,341 | 9 | CNAME | 1 | 0 | 0 |
| DS | 4,010,478 | 265 | 0 | MX, TXT, AAAA | 0 | 0 | 0 |
| TXT | 82,897 | 282 | 0 | | | | |
| GLUE6 | 13 | 0 | 0 | | | | |

**Table 8: Top 5 IP Addresses Set in BNS TLDs.**

| Handshake | | | | Decentraweb | | | |
|---|---|---|---|---|---|---|---|
| IP Address | All | BNS | ICANN | IP Address | All | BNS | ICANN |
| 44.231.6.183 | 8,980,861 | 5,272 | 5 | 147.75.40.150 | 1 | 1 | 0 |
| 54.214.136.246 | 7,686,904 | 642 | 0 | 75.2.70.75 | 1 | 1 | 0 |
| 34.123.215.203 | 44,885 | 78 | 0 | 99.83.190.102 | 1 | 1 | 0 |
| 45.79.95.228 | 966 | 7 | 0 | | | | |
| 45.79.214.114 | 843 | 29 | 0 | | | | |

**Table 9: Top 5 Domain Names Set in Handshake TLDs.**

| Domain Name | All | BNS | ICANN |
|---|---|---|---|
| ns1.example.com. | 31,968 | 9 | 0 |
| ns.superlink.me. | 29,805 | 0 | 0 |
| musk.domains. | 5,857 | 0 | 0 |
| ns1.nameboard. | 3,214 | 46 | 0 |
| ns4.registry.namebase.io. | 2,515 | 237 | 0 |

Similarly, by counting the domain name sets in the GLUE4, GLUE6, and NS records for Handshake TLDs, the number of unique domain names was 16,359,258. We present the top five domain names in Table 9. The top domain name of ns1.example.com. was used with a sample command in the Handshake document [9]. Therefore, we can assume that many users executed commands by referring to the documents. Others domain names were related to specific services or organizations. For example, superlink.me. is a service that provides Handshake domain names as digital identities and nameboard. presents the search service for Handshake TLDs. Thus, certain owners are providing new Web3 services using their Handshake TLDs. However, the ns4.registry.namebase.io. was set to many TLDs that had name collisions between BNSs; 191 of them (80.6%) were registered in 2020, suggesting that they were default values in the early years of Handshake.

**Table 10: Top 5 Owner Addresses with Highest Number of BNS TLDs.**

| Handshake | | | | Decentraweb | | | |
|---|---|---|---|---|---|---|---|
| Address | All | BNS | ICANN | Address | All | BNS | ICANN |
| hs1q...xqkh | 238,420 | 14 | 0 | 0x63...Ce73 | 1,594 | 1,247 | 0 |
| hs1q...5536 | 103,797 | 0 | 0 | 0x26...214F | 423 | 384 | 0 |
| hs1q...pfk6 | 83,663 | 2 | 0 | 0xB3...20cF | 341 | 316 | 0 |
| hs1q...v2et | 57,246 | 1 | 0 | 0x46...48E5 | 270 | 252 | 0 |
| hs1q...8rek | 56,830 | 1 | 0 | 0x63...3441 | 220 | 198 | 0 |

## 5.5 Analysis Results of Owner Addresses

The top five BNS TLD owners (owner addresses) are presented in Table 10. The number of unique owner addresses was 9,768,354 for Handshake, and 631 for Decentraweb. In both BNSs, there were owners (squatters) who exclusively owned a large number of BNS TLDs from "All" results in Table 10. This exclusive ownership was confirmed in other BNSs as well [33, 37]. It has been reported that their main purposes are financial (such as advertising and resale) and malicious (such as abuse for cyberattacks). Handshake and Decentraweb can manage the root zone of BNS TLDs, and thus have a larger namespace and greater growth potential in secondary markets than other BNSs that manage TLD+1. In these secondary markets, as SLDs under BNS TLDs that were lent as stacking are also sold in addition to buying and selling BNS TLDs, Handshake and Decentraweb TLDs are also considered as being exclusively owned, primarily for financial gains. Because the cost of Handshake TLDs tends to be low (sometimes free), we can assume that there are owner addresses with a large number of TLDs, such as bots. By contrast, because Decentraweb TLDs require a certain cost, squatters tend to acquire valuable TLDs at the pinpoint, which may result in high BNS TLD name collision rates. The BNS TLDs with name collisions with ICANN TLDs were owned by addresses other than the top five. We confirmed that four Handshake TLDs with name collisions with the ICANN TLDs were held at the same address. However, all of them were pre-reserved TLDs; therefore, they were considered as claims by their original owners. Note that because Handshake uses the unspent transaction output (UTXO) model of the blockchain rather than the account model, multiple owner addresses may be held by the same user. Although we cannot identify the specific user behind the owner addresses, the TLD ownership per user may be higher than the percentage in Table 10.

# 6 DISCUSSION

## 6.1 Countermeasures Against BNS TLD Name Collisions

BNSs preliminarily restrict to register existing ICANN TLDs. However, based on the New gTLD Program in the future, it is easy to imagine that name collisions will occur between the new ICANN and BNS TLDs, as described above. Any intervention against the owners of BNS TLDs would be outside the philosophy of decentralized management. Therefore, countermeasures could be taken to alert these owners to the risk of name collisions and encourage them to apply for CI or alert users who are considering registering such BNS TLDs in the marketplaces. In particular, based on the situation of reverting to centralized management, we believe that there is a significant scope for countermeasures by marketplaces. Countermeasures by blockchain forks and/or resolvers can provide support for new ICANN TLDs; however, they require discussion and consensus by the BNS community [1]. We will keep a close eye on future developments.

Preventive countermeasures before registration would be effective for name collisions among BNS TLDs. We can increase the probability of avoiding name collisions by checking whether the TLD is considered for registration collisions with existing BNS TLDs and by registering TLDs with a combination of alphanumeric characters and emojis, as described in Section 5.2. It would also be effective for marketplaces and communities to widely inform BNS users about the problems and risks of name collisions.

## 6.2 Impact on BNS TLD Applications

In addition to domain names, BNS TLDs can be used for digital identities, wallet address aliases, Web3 SNS account names, and destination addresses for messaging tools. For example, a person domain name, which is registered by many users, as mentioned in Section 5.2, can be used as a digital identity to manage attribute information on people, organizations, devices, services, etc. in the digital space. Although the use of BNS TLDs as digital identities is beneficial for establishing personal identities and branding in the digital space, it also increases the security risks of misdirected money transfers, misdirected data transfers, and spoofing owing to BNS TLD name collisions. In addition, the registration of person domain names such as "1.firstname.lastname" and "2.firstname.lastname" as ad hoc countermeasures exacerbates these problems.

For risk mitigation, service providers utilizing BNS TLDs are required to explain the risks, precautions, and impact of the use of BNS TLDs, registration of domain names, and improvement of user understanding. When sending money and data using BNS TLDs, eKYC would be effective, using not only the verification of the BNS TLDs but also the verification of other attribute data, such as date of birth, gender, and area of residence. In contrast, BNS users can mitigate risks caused by name collisions by specifying the BNS service name of their owned TLDs.

## 6.3 Limitation

In our investigation, we did not analyze the details of SLDs under BNS TLDs. To analyze the SLDs, we must obtain the zone files of the name servers of the TLDs. Although certain registrars publish the number of SLDs under each BNS TLD, the investigations are limited because zone files are not public information.

We described the investigation results of snapshot data as of August 2023 for the two BNSs, which may differ from those of other BNSs and future services. As the BNS is still in its developmental stage, both technologically and socially, continued observational investigations are needed in the future.

# 7 CONCLUSION

Emerging technologies such as blockchain and Web3 are developing rapidly, and new technologies are gradually being introduced into the Internet backbone. However, the introduction of new technologies that replace the design and mechanisms of existing technologies are expected to result in new security issues, because the assumed trust anchors will no longer function. In this study, we investigated the new problem of BNS TLD name collisions by analyzing BNSs that enable the decentralized management of namespaces on the Internet and blockchain. Our investigation identified 6,973 name collisions between 11,042,189 Handshake and 8,134 Decentraweb TLDs. In addition, name collisions were identified against 10 operational ICANN and 3 ICANN TLD candidates in the review. Because the BNS is still in its developmental stage and the next application round of the New gTLD Program is scheduled for the future, continued investigations of BNS TLDs are future works.

For the further development and popularization of BNSs (i.e., gain more BNS users), we believe that it is essential to build a mechanism to coexist with the existing Internet by implementing Internet technologies that are compatible with the request for comments (RFCs) and collaborating with organizations and developers in charge of standardization, such as the IETF and W3C. In addition, we believe that BNS utilization will be accelerated by playing a new role; for example, because ICANN has processes for the retirement of TLDs after delegation, BNSs are expected to take on the role of a backup/mirror for the retirement of ICANN TLDs. We hope that the results of this study will contribute to the further development and spread of BNSs.

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

## A FEATURE COMPARISON AMONG BNSS

We compared Namecoin, Emercoin, ENS, Handshake, Unstoppable Domains, and Decentraweb in terms of the launch year, blockchain, TLD examples, management range of namespaces, metadata, DNS resource records, and other records in Table 11.

## B DISTRIBUTION OF HANDSHAKE TLD PRICES

The distribution of auction prices for Handshake TLDs is shown in Fig. 2. Most Handshake TLDs were registered for free, although the registration cost depended on the auction. Because most of these free TLDs were meaningless and/or long string, they were registered for free with only one bidder.

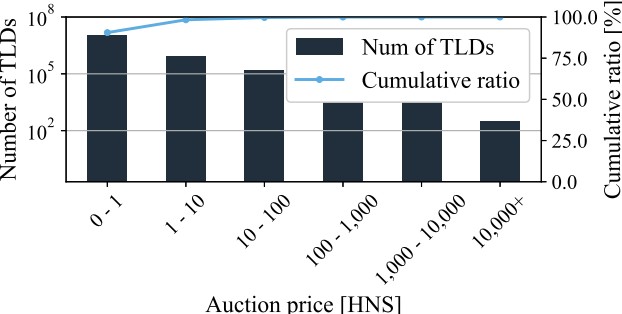

**Figure 2: Distribution of Auction Prices for Handshake TLDs.**

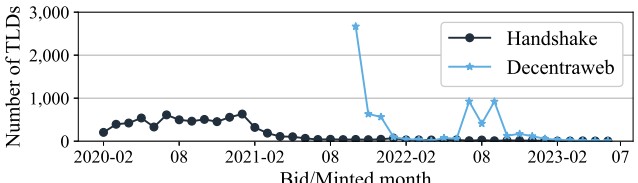

**Figure 3: Bid, Claim or Mint Date of BNS TLDs with Name Collisions.**

## C DISTRIBUTION OF TLD REGISTRATION DATES

The distribution of TLD registration dates is shown in Fig. 3. Handshake and Decentraweb made their first transactions on 4 February, 2020, and 8 October, 2021, respectively, and both services had a high number of name collisions in BNS TLDs registered several months after the service launch. In addition, Decentraweb recorded a second peak around July, 2022 when staking support for TLDs was launched.

**Table 11: Features of Each Blockchain Naming Service.**

| Service Name | Namecoin | Emercoin | ENS | Handshake | Unstoppable Domains | Decentraweb |
|---|---|---|---|---|---|---|
| Launch | 2011 | 2013 | 2017 | 2018 | 2019 | 2021 |
| Blockchain | Bitcoin fork | Bitcoin fork | Ethereum | Bcoin fork | Ethereum, Polygon | Ethereum, Polygon |
| TLD | `.bit` | `.coin, emc, .lib, .bazar` | `.eth` | Unrestricted | `.x, .crypto, .nft, .wallet` etc. | Unrestricted |
| Namespace | TLD+1 | TLD+1 | TLD+1 | TLD (Root Zone) | TLD+1 | TLD (Root Zone) |
| Metadata | Owner Address, Expiration Date etc. | Owner Address, Expiration Date etc. | Owner Address, Expiration Date etc. | Owner Address, Auction Status | Owner Address, Expiration Date etc. | Owner Address, Expiration Date etc. |
| DNS RR | RFC Compliant Records | A, AAAA, NS, TXT, PTR, CNAME, MX, SD | N/A | NS, DS, TXT, `GLUE4/6, SYNTH4/6` | RFC Compliant Records | A, AAAA, MX, CNAME, TXT |
| Other Record | URL | N/A | Contact, URL, SNS Accounts | N/A | Contact, URL SNS Accounts | Contact, URL, SNS Accounts |

