# OpenReview forum: "Investigations of Top-Level Domain Name Collisions in Blockchain Naming Services"
_ACM.org/TheWebConf/2024/Conference — TheWebConf24 Oral_

### Official Review · Reviewer_TKk2 · 2023-11-06

**Novelty:** 4
**Technical Quality:** 3

**Review:**

### Summary

In this paper, the authors conduct an empirical study about the domain name collisions in blockchain naming services. After collecting necessary information from BNS and ICANN, the authors get some interesting findings, like the characteristics of collided TLD names, the corresponding resource records, and the distribution of the owner addresses.

### Strength

- This paper investigates an interesting topic, i.e., name collisions in blockchain naming service, which is underestimated by the community currently.
- This paper delivers some interesting preliminary insights, like the existence of name collisions, the centralized management of such a decentralized domain name service, and the limited application of such a blockchain naming service.

### Weakness

- The revealed insights are not deep enough, and the adopted methods are too simple. Basically, current insights can all be concluded through some basic data processes.
- The paper is a little bit unreadable, and some notations are not introduced properly, leading to the self-contained issue. For example, in introducing DNS, the authors should also give a brief introduction of what is a resource record.

### Comments

Except for the two main concerns I raised above, here are some minor concerns.

The authors conduct empirical studies on Handshake and Decentraweb. However, there exist other blockchain naming service providers. Why choosing these two should be clarified.

At L252, the authors claim Handshake uses a fork from Bcoin. However, Bcoin is not a widely-adopted notation or a well-known blockchain platform, the authors should clarify this.

In Section 4.2.1, the authors claim that “we identified TLDs that exactly matched the names of famous organizations, corporations, brands, and web services”. However, in Section 3.2.1 and Section 3.2.2, the authors say that the registration should avoid famous companies and brands that are listed in Alexa. So, how is this possible?

At L462, the authors have collected data located in a two week period. Why choose these 14 days? Moreover, this description here is vague and unclear. The BNS data should include Handshake related data in my opinion, so why collect another bunch of Handshake data in the following? This data collecting part should be revised and clarified.

At L503, the authors claim that “However, Decentraweb users … was low”. I don’t think this is a reasonable explanation. Because at L295, the authors say registering a TLD requires at least 50 USD. Compared to the transaction fee, I don’t think the owners would not spend another piece of money to set up the resource records. I think this part should be clarified. Moreover, in Table 2, we can see that there are only 3 and 74 pieces of TLDs in Decentraweb that have DNS RRs and other records, respectively. I am curious about the reasons behind such small numbers. And I doubt the representative of Decentraweb.

In Table 3, all these notations are not well-defined. It is hard to understand for readers with a little domain knowledge.

Some descriptions are not clear. For example, the statements in Section 4.2.3. The sentence at L565, “Since 2023, … collisions occurred continuously”. And the sentence at L620, “We identified name collisions … 10 unique TLDs”.

Some typos also exist:

- L14: such decentralized management -> such a decentralized management;
- L81: NFT should be defined as non-fungible token;
- L310: metadata include the owner -> metadata includes the owner.

**Questions:**

Please refer to the above `Review` part.

**Ethics Review Description:**

-

**Reviewer Confidence:**

3: The reviewer is confident but not certain that the evaluation is correct

**Scope:**

3: The work is somewhat relevant to the Web and to the track, and is of narrow interest to a sub-community

---

### Official Review · Reviewer_9LjN · 2023-11-06

**Novelty:** 6
**Technical Quality:** 6

**Review:**

### Summary:
The paper conducts a study to look into domain name collisions both between blockchain naming services (BNSs) and traditional ICANN TLDs and among BNS services. Since BNSs build on blockchain technology, analyzing TLD-related data is a quite straightforward approach and promises to provide a complete overview. The issue of domain name collisions will only become more relevant in the future since (i) additional generic TLDs will be assigned by ICANN and (ii) the popularity of BNSs could potentially increase. Given that domain name collisions can have negative consequences, especially in light of Web3 and the transfer of funds, looking into this topic is an important matter.


### Pros:
+1: Very interesting, timely, and novel research topic

+2: Detailed analysis with lots of background information and explanations

### Cons:
-1: No mention of research artifacts or longitudinal updates

-2: Butterfly protocol is not part of the paper/study

The paper is a really interesting read, presents the information in a concise and easy-to-understand manner, and covers a previously uncovered topic. While I am also listing a few issues here, the overall quality of the paper is already quite good. Thus, please take these comments as a means to further improve your paper/work.


### Detailed Comments:

#### -1: Research Artifacts
Even though the paper presents a lot of diverse information to give a good overview of the topic at hand, the collected data probably contains even more gems. Unfortunately, the authors do not state whether they are planning to open-source their research artifacts, both in terms of software and data artifacts. In addition to allowing other researchers to go through the raw data, having access to the software/code could enable and ease follow-up research.
On a slightly different note, I would like to know whether the authors are planning to continuously update their analysis and data over time, for example, by automatically publishing new "measurements" and/or data on a dedicated website.

Both of these means could enable interesting longitudinal studies. Thus, I look forward to hearing whether this matter is on the authors' plate.


#### -2: Butterfly Protocol
In the current version, the paper considers two services for its in-depth analysis, namely, Handshake and Decentraweb. The paper also conveys why other services, such as Namecoin, Emercoin, ENS, or Unstoppable Domains, are not being considered as part of this research. However, I am surprised to discover that the Butterfly protocol (https://www.butterflyprotocol.io/) is not even mentioned in the paper. To the best of my understanding, this service is quite related to the selected ones. Consequently, at the very least, I would like to see a mention of this BNS in the main body of the paper as well as an extension of Table 11. Please educate me if I am missing something here.
That being said, I believe that the paper has significant contribution in its current form. Hence, I do not believe that the authors need to extend their analysis to include the Butterfly protocol (even if appropriate).


#### Other:
- The paper states in Section 3.2.1 and Table 11 that Handshake "uses Bcoin fork". This information appears to be incorrect since Bcoin is only a client and not a blockchain. Looking at the technical outline (https://hsd-dev.org/files/handshake.txt) seems to confirm this aspect. I believe that this aspect must be corrected.
- Would it be possible to prepare a list of TLDs that you would expect as new gTLDs for the appendix of the paper (cf. end of Section 3.3)?
- I believe that the following sentence in Section 5.4 is not correct "[...] top five IPv4 addresses set in the DNS RRs in the same manner as the DNS RRs". In any case, I am not able to grasp what the authors are trying to convey.
- The authors speculate that 191 NS records for Handshake TLDs possibly follow from the previously set default values. I am wondering whether the authors tried to look through the services' GitHub history or tried to contact the developers to confirm this hunch.
- The last sentence of the main body of the paper is quite political, in my view. Why would the authors like to see the spread of BNSs in the future? Without specifying this matter, i.e., giving a concrete reason, I would recommend the authors to omit such wording from the paper. The first part of the sentence is not affected.
- Table 11 is a nice addition to the paper and compares the different BNSs in a compact way. However, I am wondering whether the authors could extend this overview with the following properties: (a) Who operates the (underlying) blockchain of each service, (b) how many nodes are involved in the operation of the blockchain (at the time of writing), and (c) how decentralized are the nodes (in terms of operators).

#### Nits:
- Introduction: NFTs is written as "non-falsifiable tokens" in the paper; shouldn't it be "non-fungible tokens" instead?
- Related Work: Placing a comma after "best of our knowledge" would improve the readability.
- Background: Placing commas around "compliant with the ERC-721 standard" and after "SNS account IDs" would improve the readability.
- Investigation Results: Section 5.1 contains a broken sentence, which starts with "However, only 78 TLDs (1.0". Moreover, is "By contrast" in Section 5.2 correct? I would have rather used "In contrast".

### Post-Rebuttal

I kindly thank the authors for responding to the reviews and outlining their proposed changes.
Moreover, I am happy that we were able to resolve a misconception during the discussion period.
The path forward for the paper looks promising and I still would like to see the results published, despite the lack of discussing the implications of the reported findings in more detail.

**Questions:**

Is there a reason for not considering the Butterfly Protocol (https://www.butterflyprotocol.io/) both in the paper and the conducted study?

**Reviewer Confidence:**

3: The reviewer is confident but not certain that the evaluation is correct

**Scope:**

3: The work is somewhat relevant to the Web and to the track, and is of narrow interest to a sub-community

---

### Official Review · Reviewer_eUic · 2023-11-17

**Novelty:** 5
**Technical Quality:** 5

**Review:**

**Paper summary**

The paper investigates the security implications of decentralized management through Blockchain Naming Services (BNS) for Top-Level Domains (TLDs). In particular, it focuses on TLD name collisions. It collects and analyzes TLD registration data from two BNS platforms, i.e., Handshake and Decentraweb, and reveals significant challenges associated with BNS TLD name collisions.

**Strengths**

+ A new exploration on the TLD name collisions in BNS, and a necessary complementary of the studies on the collision problem in traditional ICANN

+ A large-scale analysis. More than 11 million registrations are analyzed.

+ Meaningful findings. The paper detects around 7,000 BNS TLD name collisions.

+ The paper is well-structured and well-written

**Weaknesses**

- The selection of the two BNS platforms needs to be justified

- The completeness of the data collection needs to be elaborated on

- The domain name squatting portion is a bit confusing

- Impact of the findings could be explored further

- Ethical consideration needs to be made clearer

**Detailed comments**

This paper conducts an investigation on the security implications of decentralized management through BNS for TLDs. It is a meaningful and practical study. The paper is well-written. Below I elaborate on the weaknesses listed above.

There should be a justification for selecting Handshake and Decentraweb as the BNS platforms for analysis. For a reader who is unfamiliar with this domain, the rationale behind this selection is not clearly articulated, raising questions about the representativeness of these platforms in the broader landscape of BNS solutions.

From the description on the data collection (Section 4.1.2), it is hard to assess the completeness of the data collection process. It would be good to provide an elaboration or an analysis on the completeness of the collected data to enhance the reliability of the study.

I am a bit confused by the domain name squatting analysis in Section 4.2.1. Why is it a critical component in the TLD collision problem?

The paper falls short in exploring the potential impact of the findings on the broader landscape of Internet governance and domain management. Even though domain names have collision on the TLD, the whole domain name may still differ. A more comprehensive discussion or data on how the identified collisions could affect the domain management is needed. It would be even better if real-world cases can be provided.

The paper should include a clear articulation of responsible disclosure. As the name collision can have far-reaching consequences, it is essential to responsibly disclose the findings to the stakeholders.

The sentence in line 512-514 is broken.

**Questions:**

Please refer to the comments above

**Ethics Review Description:**

I would request the authors to responsibly disclose the identified TLD collision to the stakeholders.

**Ethics Review Flag:**

Yes

**Reviewer Confidence:**

3: The reviewer is confident but not certain that the evaluation is correct

**Scope:**

4: The work is relevant to the Web and to the track, and is of broad interest to the community

---

### Official Review · Reviewer_Y26W · 2023-11-19

**Novelty:** 4
**Technical Quality:** 4

**Review:**

This paper investigates the top-level domain name collisions in blockchain naming services. The authors collected a large-scale dataset from two BNSs and identified existing BNS TLD name collisions.

pros:
1. This paper studies a practical and important issue.
2. The authors conducted detailed experiments.

cons:
1. How to verify the correctness of the results? Did the authors report the results to operators?
2. In addition to displaying results, it is better to conduct in-depth research on the causes of name collisions.
3. This paper has insufficient technical contributions. For example, in Section 6.1, technical solutions for resolving name collisions should be provided instead of simple discussions.

**Questions:**

all cons.

**Reviewer Confidence:**

3: The reviewer is confident but not certain that the evaluation is correct

**Scope:**

3: The work is somewhat relevant to the Web and to the track, and is of narrow interest to a sub-community

---

### Official Review · Reviewer_nY5H · 2023-11-23

**Novelty:** 5
**Technical Quality:** 5

**Review:**

This paper provides a quantitative analysis of TLD name collisions between the canonical ICANN DNS system and alternative blockchain-based systems, namely Handshake and Decentraweb.
The paper provides a meaningful overview of a problem that can expected to grow in importance in the future with tendencies to "re-decentralize" the web.
However, the paper also has potential for improvement:

* While the writing style is generally well-readable, some parts tend to be slightly "stiff" or mildly repetitive. For instance, "Handshake's blockchain uses Bcoin fork" in Section 3.2.1. Furthermore, the "Restrictions for TLD Registration" paragraphs in Section 3.2 are very similar in phrasing.
* At multiple occasions, the paper claims or at least implies that the bidding structure for acquiring TLDs in Handshake can be free in few situations. However, Section 5 reveals that 80% of all TLDs have been acquired for free.
  This aspect also highlights where the paper's untapped potentials lie: Section 5 mostly chains results and gives some explanation, but more insights and assessments of the consequences would have been desirable. For instance, do these results imply that Handshake does not have a sustainable business model, or do the few expensive bids make up for the bulk of free TLDs?
* Similarly, the phrasing of observations actively contributes to make the results more underwhelming at times. The authors could consider swapping their observations and the numbers underpinning those observations at times, i.e., focus on giving Section 5 more of a storyline instead of "mechanically" working off one aspect after the other.
* The proposed countermeasures are limited and hidden in the "Discussion" section, which is odd in face of my previous point.
  How could the countermeasures prevent deliberate collisions, e.g., to prepare attacks outlined by the authors?
  Most notably, the countermeasures do not consider a likely unavoidable tension between ICANN and any alternative DNS: Realistically, ICANN will have precedence over other systems in case of conflicts for practicability reasons. Contrarily, this cannot align with the philosophy behind other systems emphasizing their decentralized approach. Wouldn't a countermeasure, at least to prevent collisions with ICANN, be for ICANN to name a suffix for TLDs they will never consider registering?
* Finally, the paper would profit from a more detailed technical background on how BNSs would be used by the end-users and so, by extension, who would be affected by, e.g., attacks stemming from deliberate collisions?

**Minor Remarks:**

* Section 1: It could be argued that DNS is rather part of the Web's backbone, as the Internet's backbone could be more closely associated with the physical (network) infrastructure.
* Section 1: "non-falsifiable tokens" vs. "non-fungible tokens" (in the abstract).
* Section 5.1: Broken sentence: "(1.0We ..."
* Footnote 1 is used twice

Update: I acknowledge that I have read the authors' rebuttal comments.

**Questions:**

See review text

**Ethics Review Description:**

-

**Reviewer Confidence:**

2: The reviewer is willing to defend the evaluation, but it is likely that the reviewer did not understand parts of the paper

**Scope:**

4: The work is relevant to the Web and to the track, and is of broad interest to the community

---

### Decision · Program_Chairs · 2024-01-22

**Decision:**

Accept (Oral)

**Comment:**

The paper received 5 reviews and the reviews were generally positive leaning. The authors engaged extensively with the reviewers during the discussion phase. However, some reviewers felt their responses were not addressing the core issues but rather the minor points from the reviews. Following the discussions, the reviewer recommendations were as follows: 3 recommending borderline and 2 recommending accept. I have thus settled in the middle and am recommending a weak acceptance. I believe this captures the reviews and ensuing discussions with the authors. The paper has many positives but also a few critical shortcomings.